# Altrenogest Supplementation during Early Pregnancy Improves Reproductive Outcome in Pigs

**DOI:** 10.3390/ani12141801

**Published:** 2022-07-14

**Authors:** Bruno Bracco Donatelli Muro, Ana Clara Rodrigues Oliveira, Rafaella Fernandes Carnevale, Diego Feitosa Leal, Matheus Saliba Monteiro, André Pegoraro Poor, Francisco Alves Pereira, Leury Jesus de Souza, Juliana Bonin Ferreira, Glen William Almond, Cesar Augusto Pospissil Garbossa

**Affiliations:** 1Department of Nutrition and Animal Production, School of Veterinary Medicine and Animal Science, University of São Paulo (USP), Pirassununga 13635-900, Brazil; anaclara0904@usp.br (A.C.R.O.); rafaella.carnevale@usp.br (R.F.C.); francisco.pereira@agroceres.com (F.A.P.); cgarbossa@usp.br (C.A.P.G.); 2Department of Population Health & Pathobiology, College of Veterinary Medicine, North Carolina State University (NCSU), Raleigh, NC 27607, USA; diegoleal6446@hotmail.com (D.F.L.); jboninf@ncsu.edu (J.B.F.); gwalmond@ncsu.edu (G.W.A.); 3Department of Preventive Veterinary Medicine and Animal Health, School of Veterinary Medicine and Animal Sciences, University of São Paulo (USP), São Paulo 05508-270, Brazil; matheus.saliba.monteiro@usp.br (M.S.M.); andre.poor@usp.br (A.P.P.); 4Luiz Queiroz College of Agriculture, University of São Paulo, Piracicaba 13418-900, Brazil; leury.souza@agroceres.com

**Keywords:** progesterone, gestation, birth weight, stillbirth, swine

## Abstract

**Simple Summary:**

Over the last decades, sows’ genetic selection was aimed at increasing litter size to improve profitability in the swine industry. However, large litter size has been associated with low birth weight, increased stillbirth rate and high perinatal mortality, which are traits linked to compromised welfare in modern pig production. These side effects related to large litter size are, in part, associated with early conceptuses and placental development. The action of progesterone is essential during early pregnancy stimulating the uterine tissue to produce and secrete nutrients, growth factors, ions, cytokines and other substances that together support a proper early embryo development and placentation, which in turn may result in greater litter traits at birth. The present study demonstrated that altrenogest (progesterone analogue) supplementation during early pregnancy (from day 6 to 12 of pregnancy) may contribute to profitability and welfare in pig production systems since it increased the number of total piglets born and born alive and decreased the stillbirth rate and the number of low birth weight (<800 g) piglets.

**Abstract:**

Progesterone plays an important role in initial conceptus development and in a successful pregnancy, but results related to progesterone or its analogues (altrenogest) supplementation in early pregnancy of pigs are conflicting. The present study evaluated the effects of altrenogest supplementation in sows during days 6 and 12 of pregnancy on reproductive performance. On day 6 of pregnancy, 301 females were allocated at random to one of the following treatments: CON (Control: non-supplemented females, *n* = 163) or ALT (females daily supplemented with 20 mg of altrenogest, orally, from day 6 to 12 of pregnancy, *n* = 138). Ovulation was considered as occurred at 48 h after the first estrus detection to standardize the first day of pregnancy. The supplementation increased the number of total piglets born (ALT: 17.3 ± 0.4; CON: 16.6 ± 0.4), piglets born alive (ALT: 15.6 ± 0.4; CON: 14.8 ± 0.3), and placenta weight (ALT: 4.2 ± 0.1; CON: 3.8 ± 0.1) and decreased the stillbirth rate (ALT: 5.9 ± 0.6; CON: 7.6 ± 0.6) and the number of piglets born weighing less than 800 g (ALT: 6.6 ± 0.6; CON: 8.0 ± 0.6), without impairment on farrowing rate. These results demonstrated that altrenogest supplementation on swine females between days 6 and 12 of pregnancy may be used to improve reproductive performance.

## 1. Introduction

Over the last decades, pig breeding programmes aimed for a bigger litter size to increase the profitability of pork production systems, as the piglet is the final product of this chain [1]. Sows ovulated on average 15 oocytes in the nineties [2], and this ovulation rate has increased by 65–100% nowadays [1]. Hyperprolific modern females can have ovulation rates between 25 and 30 oocytes. However, the number of total piglets born was not augmented in the same proportion. Approximately 30 to 50% of the ova released from the ovary do not survive throughout gestation [3]. Thus, together with higher ovulation rates, this intense selection led to a disproportional increase in prenatal mortality [4,5], mostly due to an increase in both pre- and post-implantation loss [3,4].

The pre-implantation loss, which includes the period prior to 16 days of pregnancy [6], has been associated with embryonic heterogeneity within a litter while the post-implantation loss, after day 16 of pregnancy, has been linked to the competition for space and/or nutrients in the uterine surface and compromised placental development [1,7]. These events in early pregnancy associated with a limited uterine capacity may lead to inadequate placental surface area with the endometrium, especially of the less developed embryos, and/or improper placental angiogenesis [3]. These mechanisms impair gas and nutrient exchange between the conceptuses and endometrium resulting in lower average piglet birth weight, increased variation in piglet birth weight, and an increased proportion of piglets with low birth weight in sows with a great litter size [8]. Piglets with low birth weight are more prone to die during the farrowing process (stillbirth) and during lactation. These losses represent serious economic and welfare concerns. In addition, the low birth weight piglets have impaired post-weaning performance in comparison to heavier piglets, resulting in increased days to market [9].

Conceptuses’ elongation and implantation that take place between days 8 and 16 of pregnancy, demand significant amounts of energy [6]. During this period, the conceptus metabolism depends upon the limited amount of nutrients secreted into the uterine lumen by epithelial cells [10]. Progesterone plays a crucial role in early embryo development stimulating the production and release of uterine secretions, which provide nutrients, cytokines, growth factors, transport proteins and enzymes for conceptus development and placentation [10,11,12,13,14]. In addition, multiple studies demonstrated the benefits of progesterone supplementation in early pregnancy on ruminants [15,16,17,18]. Carter et al. [11] increased the peripheral progesterone concentration (approximately 3-fold higher) in cattle from day 3 to 7 of pregnancy using intravaginal progesterone implant and found larger conceptus on days 13 and 16 of pregnancy compared to non-treated animals. Using a similar progesterone implant from day 5 to 9 of pregnancy in cows, Mann et al. [16] demonstrated that an elevation from 3.4 to 6.1 ng/mL on peripheral progesterone concentration resulted in 4-fold larger embryos on day 16 of pregnancy. Higher progesterone concentration is also associated with both increased embryo survival and pregnancy rate in dairy cows [19,20]. These effects are associated with progesterone—induced modifications in gene expression in uterine tissue, thereby resulting in greater availability of nutrients and growth factors to support embryo development prior to implantation [21,22]. However, the results regarding the use of exogenous progesterone or progestogens in swine during early pregnancy are conflicting [23,24,25,26].

Untimely or prolonged exposure of the endometrium to high concentrations of progesterone or its analogues may impair early embryo development and placentation in pigs by down-regulating progesterone receptor expression in glandular and luminal uterine epithelium. Progesterone or its analogues supplementation prior to day 4 of sows’ pregnancy decreased embryonic survival [23], fertilization rate, pregnancy rate and litter size [24]. Likewise, primiparous sows that received progesterone therapy (2 mg of P4/kg^.75^ i.m.) every 12 h from 36 to 96 h after onset of standing estrus had a lower total number of embryos, embryo survival rate, number of viable embryos, and viable embryonal survival rate on day 28 of pregnancy [23]. However, recent studies suggested that supplementation of pregnant gilts with progesterone from day 3 to 10 of pregnancy may increase total protein content in the uterine lumen and stimulate glandular epithelium development and endometrial expression of genes essential for vascular function at day 12 of pregnancy [25]. Muro et al. [26] demonstrated that altrenogest supplementation from days 6 to 12 of pregnancy increased the embryo development at day 28 of pregnancy in sows, although the effect on litter performance at birth was not evaluated. Therefore, the present study was undertaken to evaluate the effects of altrenogest supplementation (Regumate^®^) in sows from day 6 to 12 of pregnancy on the reproductive performance.

## 2. Materials and Methods

### 2.1. Animals, Housing and Management

The study was performed from June 2020 until July 2021 at the experimental swine unit of Agroceres Multimix, located in Patrocínio, Minas Gerais, Brazil. A total of 301 hybrid commercial females (Landrace × Large White, PIC Cambrough 25^®^, Patos de Minas, MG, Brazil) with parity ranging from 1 to 8 were used. The average parity was 3.5 ± 0.1 (ALT: 3.2 ± 0.2; CON: 3.4 ± 0.2). The animals were checked for signs of estrus twice daily (08h00 and 16h00) by fence-line contact with a mature boar and a back-pressure test. All females were artificially inseminated with previously evaluated semen obtained from hybrid boars with proven fertility (PIC AGPIC 337^®^, Patos de Minas, MG, Brazil). The animals were inseminated at the first signs of estrus and every 24 h while the females showed standing estrus reflex. All animals were allocated in individual pens during the initial 80 days of pregnancy and then transferred to collective pens in the final 30 days of pregnancy. The females were transferred to the farrowing room one week prior to the expected farrowing date. All animals were submitted to similar nutritional and sanitary management.

### 2.2. Experimental Design

On day 6 of pregnancy the females, considered as the experimental unit, were randomly allocated to one of the following treatments: CON (non-supplemented females, *n* = 163) or ALT (females supplemented daily with 20 mg of altrenogest (Regumate^®^ MSD Animal Health, São Paulo, Brazil), orally, from day 6 to 12 of pregnancy, *n* = 138) (Figure 1). The altrenogest was provided on top of the feed and the animals were monitored until the feed was completely ingested. Ovulation was considered to occur at 48 h after the first estrus detection in order to standardize the first day of pregnancy, as shown in Figure 1.

### 2.3. Data Collection

Total piglets born, born alive, stillborn and mummified piglets were measured at birth. Stillborn and mummified piglets are presented as a percentage of total piglets born. All the piglets were individually weighed at birth to collect the data regarding average birth weight, total litter weight, percentage of piglets born with body weight under 800 g, and coefficient of variation of piglets’ weight. Both live-born and stillborn piglets were used to calculate the average birth weight. Total litter weight was considered as the sum of the body weight of live-born and stillborn piglets. Every piece of placentae that was expelled during farrowing was weighed together at the end of farrowing to collect data referring to placental weight. The farrowing rate was calculated by dividing the number of females that farrowed by the number of females inseminated. All the measurements were assessed at farrowing and the variables represent the reproductive performance at birth.

### 2.4. Statistical Analyses

Data were analyzed using generalized linear mixed models where treatments (ALT and CON) were considered fixed effects and parity was considered a random effect. The data were presented as mean ± SEM. The results were considered significant at *p* < 0.05 and marginally significant at 0.05 ≤ *p* ≤ 0.10. The software R (R Core Team, version 3.6.1, Vienna, Austria) was used.

Total born, born alive, litter weight, total placental weight, coefficient of variation of piglets’ weight and mean piglet body weight were fit using a normal distribution. Farrowing rate, percentage of piglets born with body weight under 800 g, percentage of mummies and stillborn piglets per litter were fit using a binomial distribution.

The model was adjusted for each variable using covariates when they significantly improved the model fit. The number of total piglets born was used as a covariate for total placenta weight, litter weight, coefficient of variation of piglets’ weight at birth and percentage of piglets born with body weight under 800 g.

## 3. Results

The farrowing rate was similar between groups (94.3% and 93.2% for CON and ALT, respectively; *p* > 0.10). The treatment increased (*p* < 0.05) the number of total piglets born and piglets born alive. The stillbirth rate was lower (*p* < 0.05) in ALT-females, while the percentage of mummies per litter was lower (*p* < 0.05) in CON-females.

The animals from both groups had piglets with similar (*p* > 0.10) average birth weight and litter weight; however, ALT-females had a lower percentage of piglets born with body weight under 800 g when compared to CON-females. The total placental weight from the ALT group was heavier (*p* < 0.05) than the total placental weight from the CON group. The coefficient of variation of the piglet’s birth weight was marginally lower (0.05 ≤ *p* ≤ 0.10) in the ALT group. All the results are shown in Table 1.

## 4. Discussion

In the present study, altrenogest supplementation from day 6 to 12 of pregnancy increased the number of total piglets born (0.6 piglets per female) and the number of piglets born alive (0.8 piglets per female) without affecting the farrowing rate. These results may be associated with a higher embryo and fetal survival in sows supplemented with altrenogest. Findings from Soede et al. [24] demonstrated a significantly reduced pregnancy rate, number of fetuses at day 42 of pregnancy and litter size after altrenogest supplementation. The difference between the two studies was the period of supplementation. Soede et al. [24] began the supplementation 4 days after the onset of estrus. In contrast, the present study initiated the treatment at 6 days of pregnancy. It has been demonstrated that day 6 of pregnancy can be used as a basis to initiate the supplementation of long-acting injectable progesterone or altrenogest in sows and gilts without deleterious effects on embryo survival [26]. The discrepancy among the studies may be related to the period of altrenogest (or progesterone) supplementation. The establishment of pregnancy in swine depends on the downregulation of progesterone receptors, which enables attachment of the conceptus to the uterine surface [27]. The mechanisms controlling progesterone receptor downregulation are progesterone-dependent and occur between days 3–6 of pregnancy [27]. Therefore, progesterone supplementation prior to day 6 of pregnancy may cause modifications on progesterone downregulation, and thus impair conceptuses’ development and survival.

Excessively large litter size is a major factor involved in fetal losses by mummification which is attributed to an insufficient uterine space to maintain fetal development and survival [28,29,30]. In the present study, an increase in total piglets born was accompanied by an increase of 33% of mummies in supplemented females. Similarly, Borges et al. [28] demonstrated that larger litters (more than 12 piglets/litter) had a 14.5% greater chance of presenting fetal mummification compared to litters with fewer than 10 piglets. Furthermore, intrauterine crowding and an associated decrease in endometrial surface area per fetus may lead to placental insufficiency and increased fetal mortality [30,31]. Therefore, the higher percentage of mummies in the ALT group may be due to the increase in litter size, and consequently smaller uterine space. However, the mechanisms regarding altrenogest supplementation and uterine crowding were not evaluated in the present study.

It is a consensus that increased litter size is associated with decreased placental/piglet ratio and fetal weights, which in turn predispose to increased piglets’ peri-natal mortality [32,33,34,35]. However, in the present study, altrenogest supplementation increased 0.7 total piglets born without a deleterious impact on birth weight. These results may be associated with altrenogest modulated changes in the composition of nutrients and growth factors secreted intrauterine by luminal and glandular uterine epithelium, which in turn, led to greater early conceptuses development [10,17]. In species with epitheliochorial placentation, such as pigs, the early stages of embryo development (before 25 days of pregnancy) depend on the metabolic pathways that take place in the uterine lumen; the transfer of nutrients through the vascular system is not properly developed during this period [6]. The pig conceptuses metabolize fructose and glucose as the most important nutrients to support their early development. Some amino acids such as serine, glycine, glutamate, leucine and arginine can also support early embryo development through the pentose phosphate pathway, tricarboxylic acid cycle and oxidative phosphorylation [6]. Progesterone supplementation during early pregnancy is associated with the augmented secretory activity of luminal and glandular epithelium in the ovine uterus, thereby, resulting in greater concentrations of aspartic acid, asparagine, citrulline, alanine, serine, glutamine, arginine and glucose during the pre-implantation period [18]. Moreover, progesterone supplementation in early pregnancy also increases the abundance of proteins related to the transport of nutrients to the uterine lumen during the pre-implantation period [17,36,37]. Although the modulation of intrauterine nutrients by progesterone/progestogen has not been studied in pigs, data from ewe models could be applicable, as ewe and pig conceptuses experience similar metabolic routes during early development.

Growth factors are essential in early embryo development by promoting cellular proliferation, remodelling and migration that will lead to placentation and angiogenesis [22]. Gilts supplemented with progesterone on days 3–4 of pregnancy (25 mg/100 kg) and 5–10 of pregnancy (50 mg/100 kg) had increased uterine weight, total uterine protein content and increased expression of vascular endothelial growth factor (VEGF) in the endometrium by day 12 of gestation [25]. Additionally, sows treated daily with 20 mg of altrenogest from days 6–12 of pregnancy had a higher endometrial expression of insulin-like growth factor I (IGF-I) by day 13 of pregnancy [38]. Among these molecules, IGF-1 influences conceptus growth and endometrial architecture [22], and it is positively related to fetal weight [39]. In agreement with our findings, Muro et al. [26] demonstrated that weaned sows supplemented orally with 20 mg of altrenogest from days 6 to 12 of pregnancy, had larger and heavier embryos at day 28 of pregnancy. In addition, increases in fetal weight or litter size tend to be preceded by an increase in placenta growth, which accommodates the need of the fetus during those periods of accelerated growth [33]. In fact, the placental weight was greater in the females that were supplemented with altrenogest. Therefore, the results of the present study suggest that the improvements in early embryo development may lead to increased litter size and a lower number of piglets born under 800 g. Even though the experimental design of the present study was not aimed at investigating parity or seasonal effects, additional analyses showed that there was no significant interaction between treatment and parity, nor a seasonal effect. Further studies are needed to better understand the effects of altrenogest supplementation during early pregnancy according to parity, corporal composition and under different environmental conditions.

Low birth weight is commonly reported as a risk factor for a higher stillbirth rate [40,41,42]. In the present study, sows supplemented with altrenogest during days 6 to 12 of pregnancy had a lower stillbirth rate (22% lower for ALT-sows) and a lower percentage of piglets born with body weight under 800 g (21% lower for ALT-sows) compared to non-supplemented sows. The decreased stillbirth rate found in the ALT group may be due to the lower percentage of piglets born with low birth weight (<800 g). In agreement with our findings, Fix et al. [42] observed that as birth weight increased, the likelihood of pigs being born alive also increased, which is in agreement with other authors [40,43]. The altrenogest supplementation may have led to improved early conceptus development and, consequently, a lower percentage of piglets born under 800 g [25,38]. One of the leading causes of stillbirths in pigs is asphyxia during parturition [42,44]. Low birth weight piglets may be more susceptible to asphyxia during parturition and as a result, are more prone to being stillborn [40,44]. Therefore, the decreased number of light piglets at birth may have contributed to a decreased stillbirth rate for the ALT group.

## 5. Conclusions

Oral altrenogest supplementation (20 mg/day) from day 6 to 12 of pregnancy increased the number of total piglets born, piglets born alive and placenta weight as well as decreased the stillbirth rate and the percentage of low birth weight piglets. Based on these results, it is hypothesized that oral altrenogest supplementation modulated the intrauterine environment prior to conceptuses implantation by increasing the availability of nutrients and growth factors to support embryo development and placentation. The improved early conceptuses’ growth may lead to a reduced percentage of piglets with low birth weight (<800 g) and stillbirth rate without impairments on sows’ farrowing rate. Nevertheless, further studies are required to elucidate the mechanisms of altrenogest supplementation on the uterine environment and the effects of this supplementation in sows with different characteristics (parity, genetic and corporal condition) and/or under different conditions (housing, temperature, feeding and stress).

## Figures and Tables

**Figure 1 animals-12-01801-f001:**
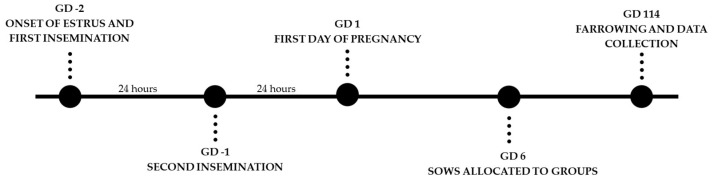
Scheme of study’s experimental design. On gestation day (GD) 6, the females were allocated at random to one of the following treatments: CON (non-supplemented females, *n* = 163) or ALT (females supplemented daily with 20 mg of altrenogest—Regumate^®^ MSD Animal Health, orally, from day 6 to 12 of pregnancy, *n* = 138).

**Table 1 animals-12-01801-t001:** Effects of altrenogest (Regumate^®^, Merck—Animal Health) supplementation from day 6 to 12 of pregnancy on sows’ and litter performance at birth.

Variable	CON ^1^	ALT ^2^	*p*-Value
Number of females (*n*)	163	138	
Parity	3.4 ± 0.2	3.2 ± 0.2	0.40
Weaning-to-estrus interval	4.3 ± 0.1	4.3 ± 0.1	0.90
Farrowing rate (%)	94.3 ± 2.61	93.2 ± 2.62	0.50
Total piglets born (*n*)	16.6 ± 0.36	17.3 ± 0.37	0.03
Born alive (*n*)	14.8 ± 0.34	15.6 ± 0.36	0.02
Stillbirth rate (%)	7.6 ± 0.58	5.9 ± 0.56	0.02
Mummies (%)	2.4 ± 0.39	3.2 ± 0.40	0.05
Total placenta weight ^4^ (kg)	3.8 ± 0.15	4.2 ± 0.15	<0.01
Total litter weight ^4^ (kg)	20.7 ± 0.45	21.3 ± 0.46	0.14
Average birth weight (kg)	1.288 ± 0.02	1.293 ± 0.02	0.80
Piglets born < 800 g ^4^ (%)	8.0 ± 0.60	6.6 ± 0.56	0.02
CV ^3^ of birth weight ^4^ (kg)	22.8 ± 1.04	21.6 ± 1.08	0.09

^1^ CON: non-treated sows; ^2^ ALT: sows treated with altrenogest (Regumate^®^, MSD—Saúde Animal, São Paulo, Brazil) from day 6 to 12 of pregnancy; ^3^ CV: coefficient of variation. Data were significantly different at *p* < 0.05 and marginally significant at 0.05 ≤ *p* ≤ 0.10; ^4^ Variables that had “total born” included as a covariate in the statistical analysis.

## Data Availability

The data presented in this study are available on request from the corresponding author.

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
