# Peer review of "Altrenogest Supplementation during Early Pregnancy Improves Reproductive Outcome in Pigs"

_animals, 2022, doi:10.3390/ani12141801_

Round 1

Reviewer 1 Report

The study entitled "Alterenogest supplementation during early pregnancy improves reproductive outcome in pigs" is overall well written. 

However, in my opinion, it presents few major flaws that need attention.

First of all, I didn't see in the manuscript any declaration of ethical clearance obtained for this study.

With regard to the experimental design it would be interesting to know why the groups are not balanced in number of animals. Furthermore the parity number of the enrolled saws seems quite wide. I would have at least removed the parity 1 from the study.

The treatment group received the Atrenogest product which, however, does not contain only the progesterone analogue but also soy oil and other excipients. Therefore, the control group should have received a placebo containing the same excipients of the treatment product to avoid any bias in the results.

The authors should clearly indicate if a placebo was used in the control group, otherwise they should justify the reason for not choosing to do so. 

Author Response

Dear Reviewer 1, we appreciate your comments and compliments to this manuscript. Please check the authors’ answers to your concerns below

Comments to Author: 

The study entitled "Alterenogest supplementation during early pregnancy improves reproductive outcome in pigs" is overall well written.

However, in my opinion, it presents few major flaws that need attention.

First of all, I didn't see in the manuscript any declaration of ethical clearance obtained for this study.

ANSWER: Dear Reviewer 1, the declaration of ethical approval is in the end of the manuscript in a section named “Institutional Review Board Statement”.

Comments to Author:

 With regard to the experimental design it would be interesting to know why the groups are not balanced in number of animals. Furthermore the parity number of the enrolled saws seems quite wide. I would have at least removed the parity 1 from the study.

ANSWER: The groups are not balanced due to a logistical problem we had with the product delivery in the last 2 months of the experiment. For this reason, there were less animals included in the treated group compared to control group in the last two batches.

The parity has a wide range because this experiment was designed to simulate commercial condition as much as possible.

We understand that differences between parities may occur, but this experiment was not designed to show it. We emphasized in the conclusion that more studies are needed to understand the effects of altrenogest treatment during early gestation according parity.

Comments to Author: 

The treatment group received the Atrenogest product which, however, does not contain only the progesterone analogue but also soy oil and other excipients. Therefore, the control group should have received a placebo containing the same excipients of the treatment product to avoid any bias in the results.

The authors should clearly indicate if a placebo was used in the control group, otherwise they should justify the reason for not choosing to do so.

ANSWER: Dear Reviewer, you raised up an interesting point. We estimated the additional energy intake between 30 – 40 kcal/day provided by the product (5 ml). It is lower than 2% of total daily energy intake of control sows. For this reason, we do not believe in any bias related to the nutritional composition of the product affecting the results of the present paper. The inclusion of placebo for control females will be considered in further trials in this topic.

Reviewer 2 Report

General

This study evaluates effects of a progesterone analogue supplementation during early pregnancy (daily dosage of 20 mg altrenogest from day 6-12 of pregnancy) on reproductive outcome in pigs. The study nicely follows up on the existing literature on the use of altrenogest and uses a good number of sows to evaluate both farrowing rate, litter size and piglet characteristics. The altrenogest treatment resulted in a higher litter size, both total born and live born, a higher mummie rate, a lower stillbirth rate, a higher total placenta weight and a lower percentage of piglets with a low birth weight (<800 g). Thus, the authors conclude that altrenogest supplementation improved reproductive performance.

The paper has several issues that need to be addressed before the paper can be accepted.  The paper needs thorough language revision for grammar, too long sentences and overall clarity. Some examples are given in the list of comments below. The introduction section is worded too generally, actually the start of the discussion (L148-155) would fit better in this introduction.  The results section only provides information on the treatment effect for all animals, but no attempt has been made to provide more information, e.g. about the parity effect (was there a general improvement or was it more or less clear in younger sows?) or the seasonal effect, or to investigate the relation between parameters (e.g. treatment decreased stillbirth rate and percentage of piglets with low birth weight, but were these related on an animal level?). As a result, also the discussion remains quite general and leaves questions open that possibly could have been answered.

Simple summary

L 17 The first sentence is much too long. It needs to be broken down in multiple sentences. Also the wording needs improvement.

L 16 Example of wording issue: Delete ‘in’ in ‘in prenatal mortality’

L 18 Example of wording issue: Change ‘on’ in  ‘in’.

L19 change order of words to ‘supplementation’ of progesterone or its analogues..

L 20 Example of wording issue: Delete ‘the’ in ‘the effects of..’

L 21 add supplementation did not affect farrowing rate, but

L 23 delete ‘and, consequently, productivity of the system. What is ‘productivity of the system’? This was not discussed and should be deleted.

Abstract

L 25 Example of wording issue: Delete ‘the’ in ‘the progesterone’.

L 26 Example of wording issue: ‘which aimed greater litter size’. Change.

L 31 Why were females ‘at random’  allocated to treatment, and not blocked for e.g. parity?

L 34-36 Add numbers (mean±sem).

L 40 Delete keyword IUGR

Introduction

The introduction is worded to generally and not always clearly referenced.

L44-46: What was the ovulation rate in earlier years and what is it now? Please provide numbers. Is it logical to refer to a paper from 1999 here?

L 46  ‘.. which impacts the returns of litter size improvement’. What do you mean?

L 47. Please add numbers. Be more specific.

L 48 Add ‘both’; an increase in both pre- and post-implantation loss.

L 49 Add time period.

L 51 ‘These aspects in early pregnancy may results..’ Please add information on the possible mechanism.

L 61. Why are these supplementations in ruminants beneficial? What is the mechanism behind it? Is it beneficial in all cattle? Or related with body condition, parity, …?

L 59 ‘and profoundly influences a successful pregnancy’. Unclear. The presence, level or profile of progesterone? When/how?

L 64. The information as provided in L148-155 is better suited here as it gives more detailed information. In this part I would expect some information on the mechanisms explaining these results.

Material and Methods

L. 80. Provide mean±SD of sow parity, lactation length, number piglets weaned.

L. 85 Was there a pregnancy check?

L 91-92. What is a completely random design? 

L 92  What do you mean with ‘the female is the experimental unit’? Did some females have more than one litter in the experiment? If so, how many?

L 93. Allocated ‘at random’. Why were they not blocked by parity and weaning-to-oestrus interval?

L. 96 ‘until the treatment was completely ingested’. Change the word treatment.

L. 106 Does average birth weight concern the live and stillborn piglets? Does total litter weight include the stillborn piglets?   ‘The placenta’ was weighed. Do you mean all individual placentae or all together? Farrowing rate was not collected. Reword.

L 115. Parity was included as random effect. Additional analyses should be done including parity class as an effect and evaluating the treatment effects per parity class. Why was season not included in the model?

L 115. Female was considered a random effect. See earlier question at L 92

L. 118-125. Each time be very specific in your wording. For example, ‘mummies’ (L 120) is this the number of mummies or the percentage of mummies per litter?

L. 125 ‘… to calculate total litter weight at birth’. Incorrect wording.

Results

Add a table showing parity effects of the treatment.

L 130 delete ‘comparing both groups’

L 134 ‘the placenta’ is not a correct term; use e.g. total placenta weight

Table 1

-      Add average parity and average weaning-to-oestrus interval for both treatment groups.

-      Add a note which covariates were used for which parameters.

Discussion

L 148 -..   Avoid repeating the earlier findings (are in introduction), only provide information on the possible causes of improved vs decreased performance in these earlier experiments, depending on the timing.

L 166-178 relates the higher mummie rate in ALT to the higher litter size in ALT. I believe you need to check this in your own data, adding mummies as a covariate and the interaction between mummies and treatment to your litter size analyses. That should clarify if and how the higher mummie rate in ALT is related to the higher littersize.

L180. Same as above for litter size and birth weight.

L 186 Explain ‘enriched’.

L. 13 Explain ‘improved’

L. 193 Delete ‘Indeed, the progesterone … establishment’. Or rephrase to clarify.

L. 197 ‘acts as a limiting factor for the final size’..  I do not understand this.

L. 200 Change ‘presented’ into ‘had’

L. 204 ‘carry-over effect to the newborn piglet. Too vague.

L. 206-211 relates the lower stillbirth rate in ALT to the lower percentage of piglets born under 800g. See remarks at L 166-178. Check this in your own data.

L. 216 ‘The decreased number … at the onset of parturition’. I do not understand this.

L. 218-221 should be moved up.

L. 222 I miss a concluding remark on the proposed mechanisms for these consequences of early altrenogest treatment.

Author Response

Dear Reviewer 2, we appreciate your contributions to this manuscript and we would like to inform you that several changes were made accordingly to your comments.

Comments to Author: 

This study evaluates effects of a progesterone analogue supplementation during early pregnancy (daily dosage of 20 mg altrenogest from day 6-12 of pregnancy) on reproductive outcome in pigs. The study nicely follows up on the existing literature on the use of altrenogest and uses a good number of sows to evaluate both farrowing rate, litter size and piglet characteristics. The altrenogest treatment resulted in a higher litter size, both total born and live born, a higher mummie rate, a lower stillbirth rate, a higher total placenta weight and a lower percentage of piglets with a low birth weight (<800 g). Thus, the authors conclude that altrenogest supplementation improved reproductive performance.

The paper has several issues that need to be addressed before the paper can be accepted.  The paper needs thorough language revision for grammar, too long sentences and overall clarity. Some examples are given in the list of comments below. The introduction section is worded too generally, actually the start of the discussion (L148-155) would fit better in this introduction.  The results section only provides information on the treatment effect for all animals, but no attempt has been made to provide more information, e.g. about the parity effect (was there a general improvement or was it more or less clear in younger sows?) or the seasonal effect, or to investigate the relation between parameters (e.g. treatment decreased stillbirth rate and percentage of piglets with low birth weight, but were these related on an animal level?). As a result, also the discussion remains quite general and leaves questions open that possibly could have been answered.

ANSWER: Dear Reviewer 2, we would like to thank you for the grammar suggestions. The modification you suggested were made and also two of the authors that are native English speakers reviewed the whole manuscript for further improvements.

Several modifications were made in the introduction and discussion section to improve the quality of the manuscript as suggested.

Comments to Author: 

Simple summary

L 17 The first sentence is much too long. It needs to be broken down in multiple sentences. Also the wording needs improvement.

L 16 Example of wording issue: Delete ‘in’ in ‘in prenatal mortality’

L 18 Example of wording issue: Change ‘on’ in  ‘in’.

L19 change order of words to ‘supplementation’ of progesterone or its analogues..

L 20 Example of wording issue: Delete ‘the’ in ‘the effects of..’

L 21 add supplementation did not affect farrowing rate, but

L 23 delete ‘and, consequently, productivity of the system. What is ‘productivity of the system’? This was not discussed and should be deleted.

ANSWER: The reviewer 3 also suggested to redesign the simple summary. Therefore, it was completely rewritten to address both reviewers concerns.

Comments to Author: 

Abstract

L 25 Example of wording issue: Delete ‘the’ in ‘the progesterone’.

L 26 Example of wording issue: ‘which aimed greater litter size’. Change.

ANSWER: The modifications were made as suggested

Comments to Author: 

L 31 Why were females ‘at random’  allocated to treatment, and not blocked for e.g. parity?

ANSWER: We chose a random design to simulate commercial conditions as much as possible. Experimental designs using parity as block and / or using parity as a fixed effect will be considered in further studies.

Comments to Author: 

L 34-36 Add numbers (mean±sem).

L 40 Delete keyword IUGR

ANSWER: mean and SEM were included for the variables cited and also the keyword IUGR was deleted as suggested

Comments to Author: 

The introduction is worded to generally and not always clearly referenced.

ANSWER: Your suggestions were used to improve the scientific quality of the introduction. We included more details from the studies referenced to avoid excessive generalization.

Comments to Author: 

L44-46: What was the ovulation rate in earlier years and what is it now? Please provide numbers. Is it logical to refer to a paper from 1990 here?

ANSWER: This statement was rewritten to make it clearer. The study from 1990 was used here to reference the ovulation rate around 15 oocytes/cycle.

Comments to Author: 

L 46  ‘.. which impacts the returns of litter size improvement’. What do you mean?

L 47. Please add numbers. Be more specific.

L 48 Add ‘both’; an increase in both pre- and post-implantation loss.

L 49 Add time period.

ANSWER: The sentences were rewritten in order to make it clearer

Comments to Author: 

L 51 ‘These aspects in early pregnancy may results..’ Please add information on the possible mechanism.

ANSWER: Possible mechanisms leading to low birth weight and birth weight variation were included

Comments to Author: 

L 61. Why are these supplementations in ruminants beneficial? What is the mechanism behind it? Is it beneficial in all cattle? Or related with body condition, parity, …?

ANSWER: Well-known mechanisms related to the positive effects of progesterone supplementation in ruminants were included as suggested.

Comments to Author: 

L 59 ‘and profoundly influences a successful pregnancy’. Unclear. The presence, level or profile of progesterone? When/how?

ANSWER:This statement “and profoundly influences a successful pregnancy” was removed from the text. We used this paragraph to explain the mechanisms by progesterone and its supplementation may influence the uterine environment and how it is related to a improved early embryo development

Comments to Author: 

L 64. The information as provided in L148-155 is better suited here as it gives more detailed information. In this part I would expect some information on the mechanisms explaining these results.

ANSWER: Part of the paragraph cited by the reviewer was transferred to the introduction.

Comments to Author: 

  1. 80. Provide mean±SD of sow parity, lactation length, number piglets weaned.

ANSWER: Mean and SEM of parity was included. However, the trial finished after farrowing and, thus, we do not have information regarding number of piglets weaned or lactation length.

Comments to Author: 

  1. 85 Was there a pregnancy check?

ANSWER: The pregnancy was checked by the farm workers by standing reflex 21 and 42 days after insemination. As this management is not the best scientific approach (ultrasound diagnosis would be the standard) and was not carried out by members of our group, we decided to do not include it in the paper and report only the farrowing rate (number of sows that farrowed divided by number of sows inseminated).

Comments to Author: 

L 91-92. What is a completely random design? 

L 92  What do you mean with ‘the female is the experimental unit’? Did some females have more than one litter in the experiment? If so, how many?

L 93. Allocated ‘at random’. Why were they not blocked by parity and weaning-to-oestrus interval?

ANSWER: We modified the statement in the text for: “females were randomly allocated…”

The statement “the female was considered the experimental unit” is used to make it clear that the product was provided individually for each sow and that the statistics were performed at sow level.

We decided for random experimental design to simulate the commercial condition as much as possible. Although we used a random design the effect of parity was adjusted using parity as a random effect in the statistical model. Different approaches will be considered for future studies in this topic.

Comments to Author: 

  1. 96 ‘until the treatment was completely ingested’. Change the word treatment.

ANSWER: It was modified as suggested.

Comments to Author: 

  1. 106 Does average birth weight concern the live and stillborn piglets? Does total litter weight include the stillborn piglets?   ‘The placenta’ was weighed. Do you mean all individual placentae or all together? Farrowing rate was not collected. Reword.

ANSWER: Average birth weight and total litter weight were calculated considering piglets born alive and stillborn piglets. This information was included in the manuscript.

All pieces of placenta were weighed together at the end of farrowing and it was considered as placental weight. This information was also included in the manuscript.

The word “collected” for Farrowing rate was removed and the calculation to obtain this variable was detailed in the manuscript.

Comments to Author: 

L 115. Parity was included as random effect. Additional analyses should be done including parity class as an effect and evaluating the treatment effects per parity class. Why was season not included in the model?

ANSWER: Firstly, we would like to thank the reviewer for the willingness to contribute with the manuscript. It was a very interesting suggestion. We would like to inform that we tested parity as a fixed effect. Interaction effect between parity and treatment was not found. There were effect of parity and treatment. However, as the effect of parity would not bring novelty to the manuscript, we decided to present only the treatment effect in the manuscript.

Season effect was tested as a fixed effect and also as a random effect and in both analysis, it was not significant and, thus, it was removed from the model.

Comments to Author: 

L 115. Female was considered a random effect. See earlier question at L 92

  1. 118-125. Each time be very specific in your wording. For example, ‘mummies’ (L 120) is this the number of mummies or the percentage of mummies per litter?
  2. 125 ‘… to calculate total litter weight at birth’. Incorrect wording.

ANSWER: All the females used had only one litter. Female was included in the model because it improved the adjust of the model measured by AIC value.

Mummies in line 120 was referring to the percentage of mummies. It was modified in this line and elsewhere it was not clear.

Comments to Author: 

Add a table showing parity effects of the treatment.

ANSWER:As discussed above , as the effect of parity would not bring novelty to the manuscript, we decided to present only the treatment effect in the manuscript. Importantly, interaction between parity and treatment was not found.

Comments to Author: 

L 130 delete ‘comparing both groups’

L 134 ‘the placenta’ is not a correct term; use e.g. total placenta weight

ANSWER:It was modified as suggested. “Total placenta weight” was used here and elsewhere this variable was cited in the manuscript.

Comments to Author: 

-      Add average parity and average weaning-to-oestrus interval for both treatment groups.

-      Add a note which covariates were used for which parameters.

ANSWER: Parity and WEI were added to the table as suggested

The description of covariates used for each model was added to the footnotes

Comments to Author: 

L 148 -..   Avoid repeating the earlier findings (are in introduction), only provide information on the possible causes of improved vs decreased performance in these earlier experiments, depending on the timing.

ANSWER: Please notice that we made several changes in the discussion to detail the possible mechanisms behind our results.

Comments to Author: 

L 166-178 relates the higher mummie rate in ALT to the higher litter size in ALT. I believe you need to check this in your own data, adding mummies as a covariate and the interaction between mummies and treatment to your litter size analyses. That should clarify if and how the higher mummie rate in ALT is related to the higher littersize.

ANSWER: The proposed statistical approach was tested. We could find effect of treatment and effect of percentage of mummies on litter size. However, interaction was not found. We do not consider the experimental design used in this study as ideal to show the relation between percentage of mummies and litter size. Additionally, this information would not bring novelty to this manuscript. Therefore, we preferred do not present this data to reinforce the reader’s attention to treatment effect as it is the main focus of this study.

Comments to Author: 

L 186 Explain ‘enriched’.

  1. 13 Explain ‘improved’

ANSWER: These words were replaced by a detailed explanation about the effects of progesterone or its analogues on the availability of nutrients and growth factors in the uterine lumen during early pregnancy.

Comments to Author: 

  1. 193 Delete ‘Indeed, the progesterone … establishment’. Or rephrase to clarify.
  2. 197 ‘acts as a limiting factor for the final size’..  I do not understand this.
  3. 200 Change ‘presented’ into ‘had’
  4. 204 ‘carry-over effect to the newborn piglet. Too vague.

ANSWER: All these sentences were rewritten.

Comments to Author: 

  1. 222 I miss a concluding remark on the proposed mechanisms for these consequences of early altrenogest treatment.

ANSWER: A proposed mechanism was included in the conclusion remarks as suggested.

Reviewer 3 Report

This is a solid study examining effects of progesterone supplementation to pigs during early pregnancy.  These kinds of studies have been published before, but as the authors note, results from different studies conflict with one another.  this study does not fix this problem, but does strengthen the idea that progesterone supplementation can be beneficiary to porcine pregnancy.

The submission is well written (except it really needs an individual who speaks English as a first language to go through and  correct some consistent errors.

It seems the Simple Summary is just a repeat of the Abstract but lacks Methods.  Perhaps the authors should redesign the Simple Summary to read more in lay terms.

The Introduction and Discussion are appropriately referenced.

Author Response

Dear Reviewer 3, we appreciate your contributions to this manuscript and we would like to inform you that changes were made in the manuscript accordingly your suggestions

Comments to Author: 

This is a solid study examining effects of progesterone supplementation to pigs during early pregnancy.  These kinds of studies have been published before, but as the authors note, results from different studies conflict with one another.  this study does not fix this problem, but does strengthen the idea that progesterone supplementation can be beneficiary to porcine pregnancy.

The submission is well written (except it really needs an individual who speaks English as a first language to go through and  correct some consistent errors.

It seems the Simple Summary is just a repeat of the Abstract but lacks Methods.  Perhaps the authors should redesign the Simple Summary to read more in lay terms.

The Introduction and Discussion are appropriately referenced.

ANSWER: Dear Reviewer, two authors who are native English speakers reviewed the manuscript and corrected the grammatical mistakes. Also, the simple summary was entirely rewritten in a lay term as suggested.

Round 2

Reviewer 2 Report

The authors have done a great job improving especially the quality of information regarding the possible mechanisms in both the Introduction and Discussion. I urge the authors to include a short discussion on the fact that even though teir experimental set-up was not aimed at investigating parity or seasaonl effects, their additional analyses showed that there was no significant parity * treatment interaction, nor a seasonal effect. I believe this is relevant.

Also, here and there, still some ‘sloppy mistakes’ are present. Please, again, check the text carefully.

Some issues:

Title spelling mistake ‘Alterenogest’

l. 20 ‘greater litter traits’ is unclear

l. 26 ‘excessive low weight’ does not seem a proper term

l. 27/8 Delete the first line ‘A disproportional.. litter’. This is not relevant here.

l. 39 ‘without impairments on sow’s reproductive performance’ is not clear. Do you mean ‘farrowing rate’? Then say so.

l. 134 – 2.3  The order of variables is not logical and wording can be easier.

l. 142  Percentage of stillborn and mummified ‘per litter’. Is that ‘of total born piglets’ (so dead + alive?), or also including mummies?

l. 149  ‘female was considered random effect’.  This is (excuse the word) nonsense. Each sow is in your dataset once, which means no variation can be attributed to it. Delete female from the random effect. Possibly R does not give a ‘fault’ here, but it is useless and inappropriate.

l. 157 and 160. Both statements refer to ‘number of total piglets born’ as covariate. If the same, why in two statements?

Table 1

-      Below table make a superscript 4 – corrected for total piglets born and add the superscript to the relevant variables in the table.

l. 180 – discussion.

In the discussion, include a section on the fact that even though the experimental set-up was not aimed at investigating parity or seasonal effects,  additional analyses showed that there was no significant parity * treatment interaction, nor a seasonal effect. I believe this is relevant.

l. 264-265 ‘The decreased number … ALT-group’. This sentence needs rephrasing. How can a lower number of light piglets result in higher vitality of the litter and then resulting in a decreased stillbirth rate..? Were the lighter piglets the stillborn ones?

l. 267 - conclusions

In the conclusion it is not clear what was the result of this study. Then. Start with those and then say ‘Based on these results, it is hypothesised..’

Author Response

Dear reviewer 2, we really appreciate your contribution to improve the scientific quality of this manuscript. We agreed with all your suggestions and the modifications were made accordingly.

Main points:

  1. 142  Percentage of stillborn and mummified ‘per litter’. Is that ‘of total born piglets’ (so dead + alive?), or also including mummies?

Answer: this statement was rewritten to make it clear that the percentage of stillborn and mummified piglets were calculated as percentage of total born (including live born, stillborn, mummies)

  1. 149  ‘female was considered random effect’.  This is (excuse the word) nonsense. Each sow is in your dataset once, which means no variation can be attributed to it. Delete female from the random effect. Possibly R does not give a ‘fault’ here, but it is useless and inappropriate.

Answer: Female was excluded from the statistical model without changes in the results.

  1. 157 and 160. Both statements refer to ‘number of total piglets born’ as covariate. If the same, why in two statements?

Answer: This section was rewritten. Please notice that we excluded total born as covariate of stillborn and mummified. As these two variables were calculated as a percentage of total born, it would be inappropriate to use total born as covariate again.

  1. 180 – discussion.

In the discussion, include a section on the fact that even though the experimental set-up was not aimed at investigating parity or seasonal effects,  additional analyses showed that there was no significant parity * treatment interaction, nor a seasonal effect. I believe this is relevant.

  1. 264-265 ‘The decreased number … ALT-group’. This sentence needs rephrasing. How can a lower number of light piglets result in higher vitality of the litter and then resulting in a decreased stillbirth rate..? Were the lighter piglets the stillborn ones?

  1. 267 - conclusions

In the conclusion it is not clear what was the result of this study. Then. Start with those and then say ‘Based on these results, it is hypothesised..’

Answer: All the modifications in the discussion and conclusion were made according to your suggestions.